# The Impact of Dynamic Emissivity–Temperature Trends on Spaceborne Data: Applications to the 2001 Mount Etna Eruption

**Nikola Rogic [1,\*], Giuseppe Bilotta [2], Gaetana Ganci [2], James O. Thompson [3,4], Annalisa Cappello [2], Hazel Rymer [1], Michael S. Ramsey [3] and Fabrizio Ferrucci [1]**

[1] School of Environment, Earth and Ecosystem Sciences, The Open University, Milton Keynes MK7 6AA, UK; hazel.rymer@open.ac.uk (H.R.); fabrizio.ferrucci@open.ac.uk (F.F.)

[2] Istituto Nazionale di Geofisica e Vulcanologia, Osservatorio Etneo, 95125 Catania, Italy; giuseppe.bilotta@ingv.it (G.B.); gaetana.ganci@ingv.it (G.G.); annalisa.cappello@ingv.it (A.C.)

[3] Department of Geology and Environmental Science, University of Pittsburgh, 4107 O'Hara Street, Pittsburgh, PA 15260, USA; james.thompson@pitt.edu (J.O.T.); mramsey@pitt.edu (M.S.R.)

[4] Department of Geosciences, Baylor University, Waco, TX 76798, USA

[\*] Correspondence: n.rogic1@open.ac.uk; Tel.: +1-(813)-593-0800

**Abstract:** Spaceborne detection and measurements of high-temperature thermal anomalies enable monitoring and forecasts of lava flow propagation. The accuracy of such thermal estimates relies on the knowledge of input parameters, such as emissivity, which notably affects computation of temperature, radiant heat flux, and subsequent analyses (e.g., effusion rate and lava flow distance to run) that rely on the accuracy of observations. To address the deficit of field and laboratory-based emissivity data for inverse and forward modelling, we measured the emissivity of 'a'a lava samples from the 2001 Mt. Etna eruption, over the wide range of temperatures (773 to 1373 K) and wavelengths (2.17 to 21.0 μm). The results show that emissivity is not only wavelength dependent, but it also increases non-linearly with cooling, revealing considerably lower values than those typically assumed for basalts. This new evidence showed the largest and smallest increase in average emissivity during cooling in the MIR and TIR regions (~30% and ~8% respectively), whereas the shorter wavelengths of the SWIR region showed a moderate increase (~15%). These results applied to spaceborne data confirm that the variable emissivity-derived radiant heat flux is greater than the constant emissivity assumption. For the differences between the radiant heat flux in the case of variable and constant emissivity, we found the median value is 0.06, whereas the 25th and the 75th percentiles are 0.014 and 0.161, respectively. This new evidence has significant impacts on the modelling of lava flow simulations, causing a dissimilarity between the two emissivity approaches of ~16% in the final area and ~7% in the maximum thickness. The multicomponent emissivity input provides means for 'best practice' scenario when accurate data required. The novel approach developed here can be used to test an improved version of existing multi-platform, multi-payload volcano monitoring systems.

**Keywords:** emissivity; FTIR; remote sensing; lava flow modelling; volcano monitoring; Mount Etna

## 1. Introduction

A variety of approaches are used to derive apparent surface temperatures from spaceborne infrared (IR) data [1,2]. Land surface temperature (LST) and land surface emissivity (LSE) are two key parameters used as model input parameters, because they are closely linked to the Earth's surface energy balance [3]. However, emissivity ($\varepsilon$) has not previously been measured across the full range of lava temperatures and relevant compositions; rather it is generally assumed to have a constant value between 1 and 0.8 for basaltic lava [1].

To address this deficit, emissivity experiments on 'a'a lava samples from the 2001 Mt. Etna (Italy) eruption were performed to establish the implications that emissivity, as a model input parameter, has for deriving lava surface temperatures and radiant heat flux.

Additionally, this study aimed to provide a physical basis for avoiding assumptions when exploiting remote sensing and modelling data for emergency management and hazard forecasting purposes.

Mt Etna is one of the most active and hazardous volcanoes in the world, well known for frequent lava flow-forming eruptions from vents situated on the volcano flanks and for its summit activity, which has been almost persistent since January 2011 [4–6]. We chose the 2001 lava flow because it has not been covered by successive lava flows. Lava flows produced during recent short-lived events (e.g., 2011), lasting from a few hours to several days, overlap, thus making spaceborne identification and/or field lava sampling difficult. Additionally, the 2001 eruption is one of the major flank eruptions that has occurred at Etna in the last two decades, and was well observed by several multispectral sensors, including the Enhanced Thematic Mapper + (ETM+) and the Moderate-resolution Imaging Spectrometer (MODIS). The 2001 eruption produced seven different fast-developing lava flows (Figure 1) in only 23 days with a total bulk volume of about 40 million m$^3$ [7]. We focused on the individual flow that emanated from the southern flank at 2100 m above sea level (LFS1), which has not been covered by successive lava flows, where we collected different lava samples (Figure 1) during a field campaign in 2017.

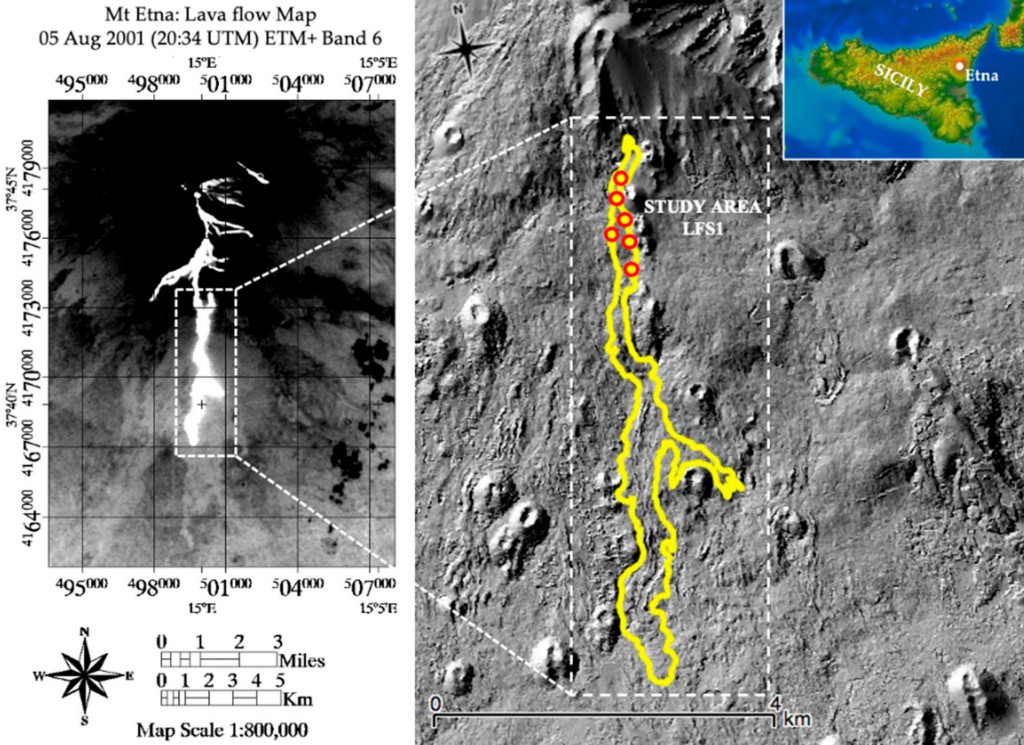

**Figure 1.** (**left**) Study area indicated on ETM+ TIR (Band 6) image, used here for visual presentation purposes alone, showing a high temperature thermal anomaly image of the 2001 Mt. Etna eruption, acquired on 5 August 2001; (**right**) the areal extent of the individual lava flow, LFS1 [7], analysed in this study, is highlighted in yellow and superimposed on a Digital Elevation Model (DEM) of Mt. Etna. Empty red circles indicate an approximate location of collected samples.

Emissivity is not well quantified for molten materials and hot volcanic rocks. Most authors adopt a constant value based on the rare, published laboratory measurements to perform thermal infrared (TIR) emissivity–temperature separation [3], underlying the Advanced Spaceborne Thermal Emission and Reflection Radiometer Global Emissivity Database (ASTER GED), among others [8]. The emissivity of a target such as that considered here (Mt. Etna) can be extracted from existing global spaceborne libraries such as the ASTER GED. However, this represents only a 'static' mean emissivity value [9]. For example, the ASTER GED 100-m pixel value is a nine-year average (2000–2008), which would integrate

(background) values outside the dimensions of the target investigated (active lava flow) due to its coarse spatial resolution. These constant emissivity values, if applied uniformly, independent of the pixel size and presence of hot material (i.e., independent of the nominal scale of observations) will produce variations in computed apparent surface temperatures and would not account for the range of temperatures (and emissivities) found in an active lava flow.

Using different emissivities (e.g., not assuming unity) acknowledges that Etnean trachy-basalts emit a percentage of radiance incident upon them. This percentage would be denoted by the wavelength-average emissivity term, and despite previously never being proven to be temperature dependent, its behaviour with temperature has been questioned [10]. Several studies of silicate glasses and basaltic lavas [11–15] suggest that the emissivity of molten material is significantly lower than that of the same material in solid state and argued to be lower than the assumed >0.80 [12,16].

In this study, laboratory measured emissivity–temperature trends provide the means of computing more accurate lava surface temperatures, which would account for both variation in emissivity with wavelength and include the range of temperatures found in an active lava flow (e.g., ≤1360 K for Mt. Etna). Satellite instruments employed in this study are MODIS, onboard NASA's Terra and Aqua satellites, and ETM+, onboard NASA/USGS's Landsat-7 satellite. Additionally, smaller ETM+ pixels will saturate more readily than larger MODIS pixels when acquiring data over an active flow (high-temperature thermal anomaly). For this reason, we used a night-time image acquired by ETM+ on 5 August 2001, as it contains only the thermally emitted ground component, which is compared with MODIS data acquired between 15 July and 30 August 2001. Please note that computation of radiant heat flux in this study, using ETM+ data, involves two SWIR bands (Band 5 and Band 7), whereas the ETM+ Band 6 (TIR) image (Figure 1) was used for visual presentation purposes alone.

A multi-sensor data approach, integrating IR observation from different spaceborne platforms, including SWIR data, has been suggested to improve information for an individual target [17] and detection of a set of targets [18]. Nonetheless, due to the low revisit rates (at least five days), and despite previous proposals [18,19], it appears that no operational system is currently actively using decametric resolution SWIR data to complement radiant heat fluxes computed in large pixels.

The ETM+ data can still provide useful 'snap-shots', indicative of the instantaneous state of activity, to compute relatively accurate radiant (and mass) flux [9]. Nonetheless, its data are based on a limited number of infrequent observations (i.e., one scene), due to the instruments' temporal resolution and meteorological conditions. Therefore, it may not reflect the significant peak discharge rate or dynamic flow regimes that are known to change frequently [20,21]. Thus, it was used here in tandem with moderate resolution MODIS data to complement computed radiant heat flux range for the period analysed.

## 2. Materials and Methods

We performed laboratory-based FTIR analyses on solidified volcanic rock samples from the 2001 eruption of Mt Etna, to derive emissivity at a range of temperatures (773–1373 K) and wavelengths (2.17–21.25 μm).

### 2.1. Emissivity from Radiance Spectra

Thermal emission spectra were collected in the Image Visualization and Infrared Spectroscopy (IVIS) Laboratory, at the University of Pittsburgh, Pennsylvania, U.S.A. The experimental setup (Figure 2) to measure absolute emissivity at very high temperatures uses a Nicolet Nexus 870 FTIR spectrometer, equipped with a potassium bromide (KBr) beam splitter, and a mercury cadmium telluride (MCT-B) detector (cooled with liquid nitrogen) with a spectral range of 4601–470 cm$^{-1}$ (2.17–21 μm). Emission spectra were collected over 8 scans (~10 s total), at a spectral resolution of 2 cm$^{-1}$ (2065 bands) and averaged to improve the signal to noise.

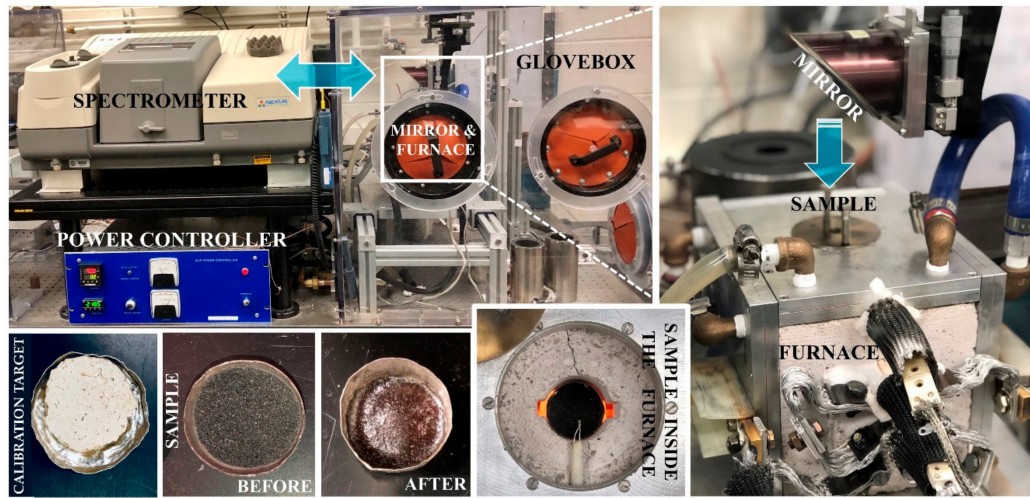

**Figure 2.** Experimental high-temperature (773–1373 K) setup to measure emissivity at IVIS Laboratory, University of Pittsburgh (USA). Shown is the power controller unit and Nicolet Nexus 870 FTIR spectrometer, adjacent to the experiment chamber (**top right**), which is continuously purged of $CO_2$ and $H_2O$. The experiment chamber contains the furnace and sample measuring apparatus (**right**). The sample before and after measurement is also shown (**bottom left**), and the calibration material (alumina) used as the blackbody source.

An experiment chamber adjacent to the spectrometer contains a custom-made furnace sample measuring apparatus [15]. The temperature and humidity of the spectrometer and experiment chamber are monitored continuously. Both the spectrometer and the attached experiment chamber are purged with dry air to limit spectral obscuration by $H_2O$ and $CO_2$. All samples were crushed and sieved into ~100–350 μm size fractions. Approximately 1 g of sample is poured into a 3.0 cm diameter platinum cup (to ~3 mm depth), which is manually placed into the furnace and covered with a furnace lid (with viewing opening) and kept there for the duration of the experiment to maintain constant conditions.

In emission spectrometry, the spectrometer measures the energy emitted from the heated sample to subsequently determine the radiance emitted from the sample surface. Sample measurement steps from 773 to 1373 K were set at 50 K intervals (e.g., 773, 823, 873, 923 K) using a power controller (Figure 2). A four-minute dwell time was applied at each temperature step, to allow equilibration prior to the collection of the spectra, which is essential for attaining accurate emissivity spectra.

Prior to any sample analysis, spectra were acquired from a blackbody calibration target with the same furnace-viewing geometry (Figure 2). Based on a well-established methodology [22], the calibration spectra were acquired at 50 K higher and lower than the expected sample temperature measurements (e.g., at 350 and 450 K for a sample temperature of 400 K). The spectra of the blackbody calibration targets allow for the instrument and environmental emission to be quantified and removed using a well-established method [22]. The use of an alumina disk as the blackbody calibration source has not been compared to a recognised national standard. However, the calibration process follows long-standing emissivity protocols [22]. The emissivity of the alumina calibration disk was measured using a Thermo Scientific iN10 FTIR microscope ($\mu$ − FTIR) collecting bi-directional reflectance spectra and converted to emissivity using Kirchhoff's law [23]. This resulted in emissivity of 0.91 across the samples spectral range with a total error of 2% [15]. The chamber temperature was monitored using a thermocouple and was recorded and applied for each temperature step. Calibration and conversion of raw data to absolute emissivity is described in previous studies [15,22]. The experimental error and uncertainty associated with the data acquired using the FTIR spectrometer with the furnace at the University of Pittsburgh's IVIS laboratory are reported to be <2% and <4%, respectively [15].

### 2.2. FTIR Data Analysis and Creation of a 'Dynamic Emissivity–Temperature Rule'

For each sample, emissivity was acquired at a range of wavelength and temperatures. Emissivity was acquired during both heating and cooling of the sample. We used the cooling data in the analysis because it more accurately represents the natural process being investigated. We aimed to derive an experimentally deduced emissivity–temperature relation in this study. The spectral radiance emitted in a wavelength interval ($\lambda_{min}$, $\lambda_{max}$) by a graybody at temperature ($T$) is:

$$\int_{\lambda_{min}}^{\lambda_{max}} \varepsilon(\lambda, T) B_\lambda(\lambda, T) d\lambda \tag{1}$$

where $B_\lambda(\lambda, T)$ is Planck's law (in wavelength). By the mean value theorem, the (spectrum-integrated) mean emissivity is:

$$\varepsilon_{[\lambda_{min}, \lambda_{max}]}(T) = \frac{\int_{\lambda_{min}}^{\lambda_{max}} \varepsilon(\lambda, T) B_\lambda(\lambda, T) d\lambda}{\int_{\lambda_{min}}^{\lambda_{max}} B_\lambda(\lambda, T) d\lambda} \tag{2}$$

and it is such that the graybody spectral radiance (in the given wavelength interval) can be computed from the blackbody spectral radiance $\int_{\lambda_{min}}^{\lambda_{max}} B_\lambda(\lambda, T) d\lambda$ by multiplying by the mean emissivity $\varepsilon_{[\lambda_{min}, \lambda_{max}]}(T)$.

Due to the discrete sampling, the information we have about $\varepsilon(\lambda, T)$ is not continuous (in $\lambda$), so for the integral we use a piecewise linear interpolation between each pair of data points, an approximation justified by the fine granularity of the wavenumber sampling in the data. A higher-order reconstruction is possible, but the results obtained provide an uncertainty of the same order as the measurement error.

For the application of satellite remote sensing, the mean emissivity was computed over the bandwidth of specific sensor channels: for MODIS, the Middle Infrared (MIR) channels 21 and 22 (3.929–3.989 µm) and the TIR channels 31 (10.780–11.280 µm) and 32 (11.770–12.270 µm); for Landsat-7, the SWIR Channel 7 (2.09–2.35 µm). Using Equation (2), we obtain one mean emissivity value per channel and per temperature, from which we derive per-channel relationships between emissivity and temperature by fitting second-degree polynomials to the computed mean emissivity values. We verify the approximation by checking that the discrepancy between the fitted values and the data is not larger than the instrument error (i.e., lower than 4%). For lava flow modelling, we are interested in the full spectrum mean emissivity (i.e., the mean emissivity across the entire spectrum). Because the data available are limited to the 2.17 to 21 µm range, we approximate the full spectrum mean emissivity by the mean emissivity over the available data range, after verifying that the mean emissivity is sufficiently 'stable' within the range. This is similar to a previous study based on multispectral ground-based TIR data [16].

### 2.3. Radiant Heat Flux from Spaceborne Data

To accurately compute the emission of energy actually leaving a surface and the radiant heat flux [24], the radiant signal is corrected for the influence of the atmospheric transfer function [25] and the emissivity of that radiating surface [1]. Here, we assessed the role of emissivity from multiplatform spaceborne data, using a multicomponent approach, instead of fixed value estimates. To achieve this, we applied a novel technique, where the computation of radiant heat fluxes from spaceborne data is based on FTIR laboratory measured data. The multicomponent emissivities are integrated by observed emissivity–temperature behaviours and synthesised in an experimentally deduced emissivity–temperature–wavelength relation.

To consider the measured emissivity–temperature trend, the HOTSAT system [26–28] was modified by designing and implementing a routine for MODIS data. The HOTSAT system locates the thermal anomalies in a defined volcanic area. The pixels detected as thermally anomalous, the 'hotspot' pixels, are non-isothermal, but they will be a mixture of

many different thermal components [1]. To provide an estimation of the radiant heat flux associated with the thermally mixed pixels, they need to be 'un-mixed'. There are different ways to solve these problems (e.g., [29]), but in this case, to consider the emissivity variations with temperature, we use a dual band model [30]. Three components are considered for each pixel, including a portion of melt at the highest temperature, a portion invaded by crusted lava, and a background (i.e., no active lava) [1]. Each of these components is solved using the emissivity–temperature relation:

$$B_{\lambda MIR}(\lambda_{MIR}, T_{int}) = pb\, \varepsilon(\lambda_{MIR}, T_b)\, B_{\lambda MIR}(\lambda_{MIR}, T_b) +$$
$$pc\, \varepsilon(\lambda_{MIR}, T_c)\, B_{\lambda MIR}(\lambda_{MIR}, T_c) + ph\, \varepsilon(\lambda_{MIR}, T_h)\, B_{\lambda MIR}(\lambda_{MIR},\, T_h) \tag{3}$$

$$(\lambda_{TIR}, T_h)\, B_{\lambda TIR}(\lambda_{TIR},\, T_h) \tag{4}$$

$$pb + pc + ph = 1 \tag{5}$$

where *pc*, *ph*, and *pb* are the pixel portions at the Tc (temperature of crusted lava), and $T_h$ and $T_b$ are the temperatures of melt lava and background, respectively. $T_{int}$ is the integrated temperature for the whole pixel. In solving the system of three equations in five unknowns ($T_h$, $T_c$, $T_b$, *pb* and *pc*), we used the 'fsolve' function implemented in MATLAB. We considered band 21 as the MIR band and 31 as the TIR band corresponding to the 3.959 and 11.03 µm central wavelengths, respectively. We assumed Th was equal to 1373 K [31], and Tb was retrieved from neighbouring pixels not affected by thermal anomalies. Thus, the system of equations was solved for $T_c$, *pb* and *pc*.

Additionally, Landsat ETM+ data and its preliminary results were used in this study for comparison purposes, which followed the specific approach [9,32] to compute radiant heat fluxes during the 2001 Mt. Etna eruption, using two ETM+ SWIR bands. This is a systemised variant of the sub-pixel resolution approaches [30,33] and their application to high-temperature volcanic features [24,34,35]. Using a 'thresholding' approach, linking a specific emissivity value to recorded radiance (Figure 3) allows the relative size and temperature of these thermal components to be resolved, following solutions, which depend on data availability (saturation) in each band [36,37]. Here, the Landsat-7 (ETM+) night-time image distributed by the Global Visualization (GloVis) Viewer [38], acquired during the 2001 Mt. Etna eruption was analysed to produce radiant heat flux values for 5 August 2001. Because emissivity also varies as a function of wavelength, the absence of laboratory FTIR data at 1.65 µm (due to instrument limitations), and the close proximity of SWIR bands, similar behavior is anticipated based on the previous research [10].

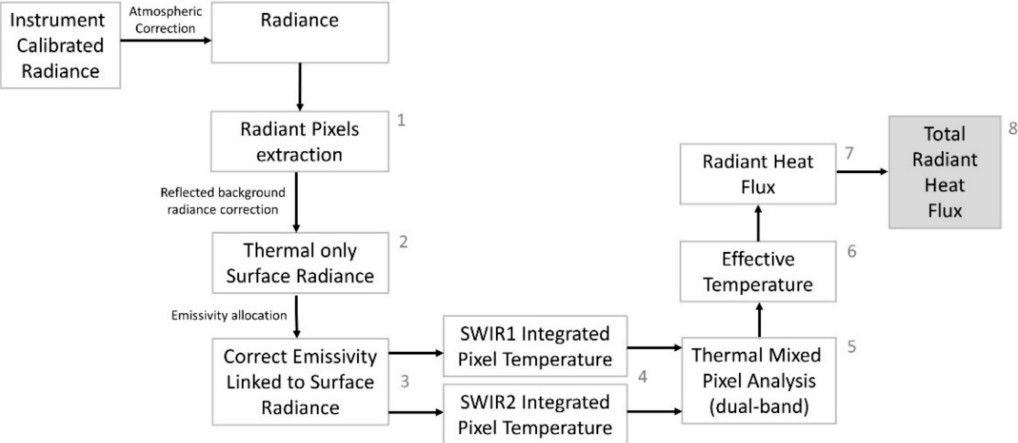

**Figure 3.** A flowchart illustrating methods (steps 1–8) to derive radiant heat flux using high-spatial resolution data (ETM+) in two SWIR bands.

## 3. Results

### 3.1. Laboratory-Based FTIR Results

FTIR emissivity was derived at a variety of sample temperatures for two trachytic basalts (NRE.4 Series) collected from lava flows emplaced during the 2001 Mt. Etna eruption. Generally, the emissivity increased as the sample temperature decreased (cooling) and a glassy crust formed. The sample temperature decreased from 1373 to 773 K, causing the average emissivity to increase from 0.7054 to 0.8647 (~23%) between 2.17 and 15 μm. The greatest and smallest increases in average emissivity were observed in the MIR region (~30%) and TIR region (~8%), respectively, with the SWIR region having a moderate increase (~15%).

Basalts have a $SiO_2$ content of 45–52% and hence have spectra that are dominated by absorption features associated with $SiO_2$ bonds (vibrations and bending) [11,39]. The strong absorption feature at ~4.0 μm (Figure 4) is a result of silica overtone vibrations, whereas the smaller feature at ~7.5 μm (Figure 4) is associated with Al-O bond vibrations. The main Si-O-Si bond vibration and bending result in the broad absorption feature between 8.0 and 12.0 μm. The increase in emissivity observed during cooling and crust formation of these samples is a consequence of the decrease in temperature. This consequently reduces $SiO_2$ bond vibrational and bending energy, reducing energy absorption by the sample. Additionally, there is a small absorption feature at ~15 μm caused by Al-OH and Si-OH bond bending; however, this is not strongly dependent on temperature fluctuations in basalts. The increase in emissivity during cooling is a result of the reduction in vibrational and bending energy of the crystal lattice within the samples. Overall, the spectral results and morphologies are dominated by the $SiO_2$ content of the basalt samples, [11,15].

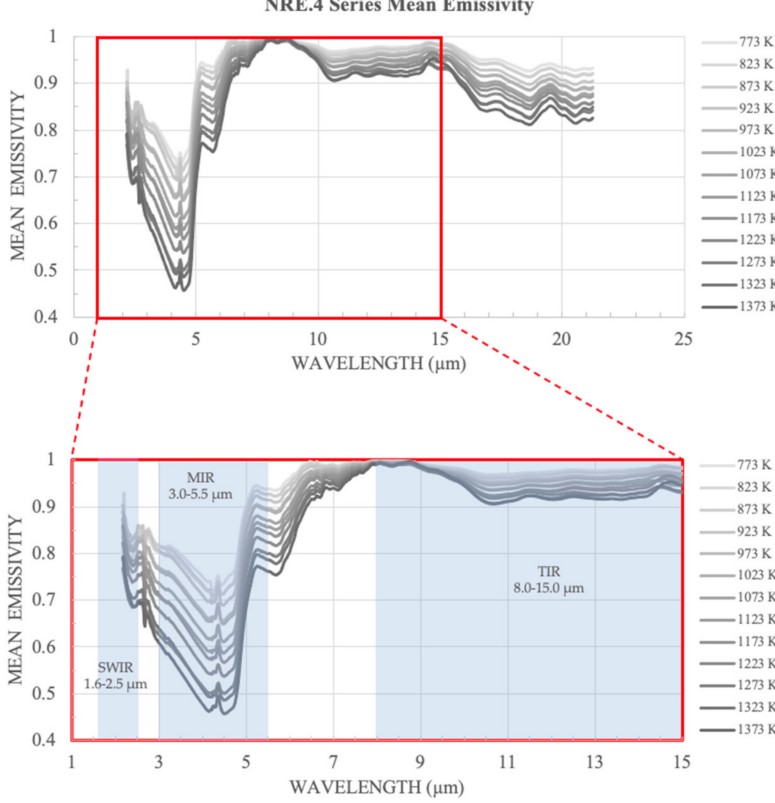

**Figure 4.** (**Top panel**) Emissivity spectra of basaltic samples, acquired using FTIR emission spectroscopy at a range of sample temperatures from 1373 to 773 K; (**bottom panel**) indicating emissivity–temperature trends in the region(s) of interest (SWIR, MIR and TIR). The samples are trachy-basaltic lavas belonging to the individual lava flow (Figure 1) emplaced between 18 July to 9 August 2001 [7]. Overall, the average emissivity increases as the temperature of the samples decreases in a nonlinear inverse relationship.

### 3.1.1. Mean Integrated Emissivity for Remote Sensing Applications

For remote sensing applications, we derive experimentally deduced emissivity–temperature relations using quadratic polynomial fitting. We do this for the SWIR Channel 7 (2.09–2.35 μm) of Landsat 7, the MODIS' MIR channels 21 and 22 (both with bandwidth 3.929–3.989 μm), and TIR channels 31 (10.780–11.280 μm) and 32 (11.770–12.270 μm).

For the Landsat 7 SWIR channel 7 (Figure 5a), the fitted quadratic polynomial is:

$$\varepsilon_{SWIR}(T) = 0.30725 + 0.00113\,T - 6.0904 \cdot 10^{-7}\,T^2 \tag{6}$$

with a maximum relative error of 0.07%, median 0.02%, and standard deviation 0.02%. For the MODIS MIR channels (Figure 5b), the fitted quadratic polynomial is:

$$\varepsilon_{MIR}(T) = 0.8559 + 0.00007\,T - 2.5241 \cdot 10^{-7}\,T^2 \tag{7}$$

with a maximum relative error of 0.04%, median 0.01%, and standard deviation 0.009%. For the TIR channels (Figure 5c,d), the fitted polynomials are:

$$\varepsilon_{TIR31}(T) = 1.0346 - 0.00007\,T - 1.2899 \cdot 10^{-8}\,T^2 \tag{8}$$

with a maximum relative error of 0.007%, median 0.003%, and standard deviation 0.002%, and:

$$\varepsilon_{TIR32}(T) = 1.0275 - 0.00004\,T - 2.6096 \cdot 10^{-8}\,T^2 \tag{9}$$

with a maximum relative error of 0.005%, median 0.001%, and standard deviation 0.001%. These are similar TIR trends as observed by a recent study [16] using field data. The fitting parameters with the associated standard error and 95% confidence intervals are reported in Table 1.

### 3.1.2. Mean Emissivity for Lava Flow Modelling

For lava flow simulations, a full spectrum mean emissivity must be computed. Since the available data only span the range from 2.17 to 21 μm, we can compute the mean emissivity in this part of the spectrum, which is assumed to be a sufficient approximation of the full spectrum emissivity for our applications. This assumption is qualitatively corroborated by the fact that, as the subset of the spectrum taken into consideration grows, the value of the mean emissivity integrated over that section of the spectrum changes less (Figure 6). To show this, consider for a given, fixed temperature $T$, the mean emissivity over two ranges, where one of the extrema is fixed (respectively at 2.17 and 21 μm) and the other is free to change. The resulting 'lower' and 'upper' functions $\varepsilon_{T,-}(\lambda) = \varepsilon_{[2.17,\lambda]}(T)$ and $\varepsilon_{T,+}(\lambda) = \varepsilon_{[\lambda,21]}(T)$ are such that $\varepsilon_{T,-}(21) = \varepsilon_{[2.17,21]}(T) = \varepsilon_{T,+}(2.17)$. This shows that a small change in the range results in a small change in the mean emissivity, i.e., that $\varepsilon_{T,-}(\lambda)$ has values close to $\varepsilon_{T,-}(21)$ for $\lambda \to 21$, and that $\varepsilon_{T,+}(\lambda)$ has values close to $\varepsilon_{T,+}(\lambda)(2.17)$ for $\lambda \to 2.17$.

Mathematically, this can be verified by looking at the magnitude of the derivatives of the two functions at the extremum of the corresponding interval (21 μm for $\varepsilon_{T,-}(\lambda)$ and 2.17 μm for $\varepsilon_{T,+}(\lambda)$). Since the functions are computed numerically from the sampled values, we cannot compute an analytical derivative. The derivatives are computed as the ratio between the change in the function values between two consecutive samples and the difference in the wavelengths. For the derivative of $\varepsilon_{T,-}(\lambda)$ at 21 μm, we take the 21 μm sample and the one before it, whereas for the derivative of $\varepsilon_{T,+}(\lambda)$ at 2.17 μm, we use the 2.17 μm sample and the one after it. For $\varepsilon_{T,-}(\lambda)$ we find that, across all sampling temperatures, the maximum derivative magnitude is $1.6 \cdot 10^{-4}$ μm$^{-1}$, whereas for $\varepsilon_{T,+}(\lambda)$ the magnitude is $4.8 \cdot 10^{-2}$ μm$^{-1}$. This suggests that the mean emissivity is very stable around $\lambda = 21$ μm. Around $\lambda = 2.17$ μm, the mean emissivity is less stable, but given the range of temperatures that can be expected from the lava and the magnitude of the derivative, we can expect an approximate change of less than 5%.

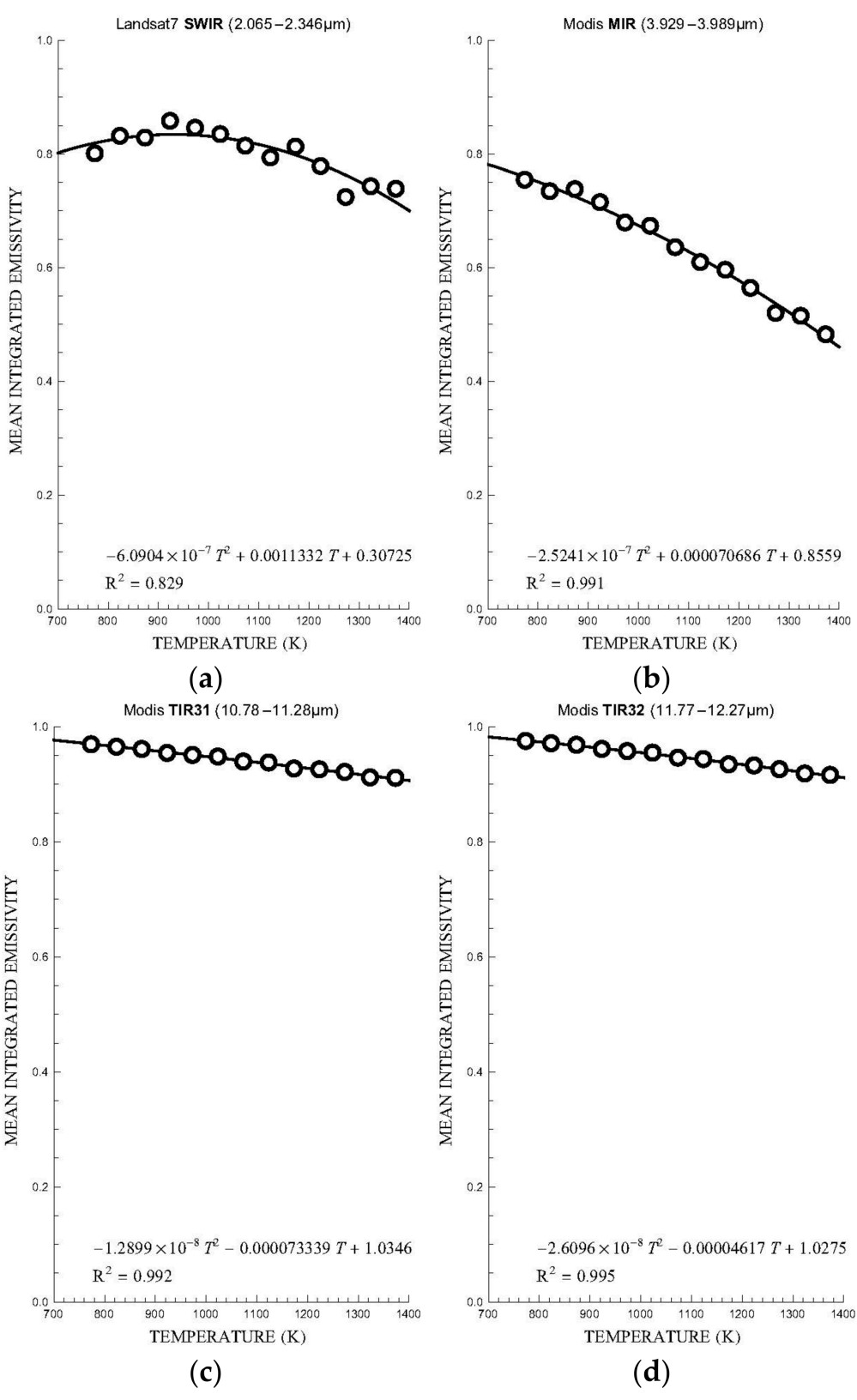

**Figure 5.** Emissivity–temperature trends at a range of temperatures (773–1373 K) in spaceborne (**a**) SWIR (ETM+) and (**b**–**d**) MIR-TIR (MODIS) bands.

**Table 1.** Fitting parameters with associated standard error and confidence intervals.

| WAVELENGTH | | ESTIMATE | STANDARD ERROR | CONFIDENCE INTERVAL |
|---|---|---|---|---|
| **SWIR** | 1 | $3.07 \times 10^{-1}$ | $1.97 \times 10^{-1}$ | $\{-3.17 \times 10^{-1}, 9.32 \times 10^{-1}\}$ |
| | T | $1.13 \times 10^{-3}$ | $3.75 \times 10^{-4}$ | $\{-5.57 \times 10^{-5}, 2.32 \times 10^{-3}\}$ |
| | $T^2$ | $-6.09 \times 10^{-7}$ | $1.74 \times 10^{-7}$ | $\{-1.16 \times 10^{-6}, -5.67 \times 10^{-8}\}$ |
| **MIR** | 1 | $8.56 \times 10^{-1}$ | $9.86 \times 10^{-2}$ | $\{5.44 \times 10^{-1}, 1.17\}$ |
| | T | $7.07 \times 10^{-5}$ | $1.88 \times 10^{-4}$ | $\{-5.24 \times 10^{-4}, 6.65 \times 10^{-4}\}$ |
| | $T^2$ | $-2.52 \times 10^{-7}$ | $8.71 \times 10^{-8}$ | $\{-5.29 \times 10^{-7}, 2.38 \times 10^{-8}\}$ |
| **TIR_31** | 1 | 1.03 | $1.98 \times 10^{-2}$ | $\{9.72 \times 10^{-1}, 1.10\}$ |
| | T | $-7.33 \times 10^{-5}$ | $3.77 \times 10^{-5}$ | $\{-1.93 \times 10^{-4}, 4.63 \times 10^{-5}\}$ |
| | $T^2$ | $-1.29 \times 10^{-8}$ | $1.75 \times 10^{-8}$ | $\{-6.85 \times 10^{-8}, 4.27 \times 10^{-8}\}$ |
| **TIR_32** | 1 | 1.03 | $1.61 \times 10^{-2}$ | $\{9.76 \times 10^{-1}, 1.08\}$ |
| | T | $-4.62 \times 10^{-5}$ | $3.07 \times 10^{-5}$ | $\{-1.44 \times 10^{-4}, 5.12 \times 10^{-5}\}$ |
| | $T^2$ | $-2.61 \times 10^{-8}$ | $1.43 \times 10^{-8}$ | $\{-7.13 \times 10^{-8}, 1.91 \times 10^{-8}\}$ |

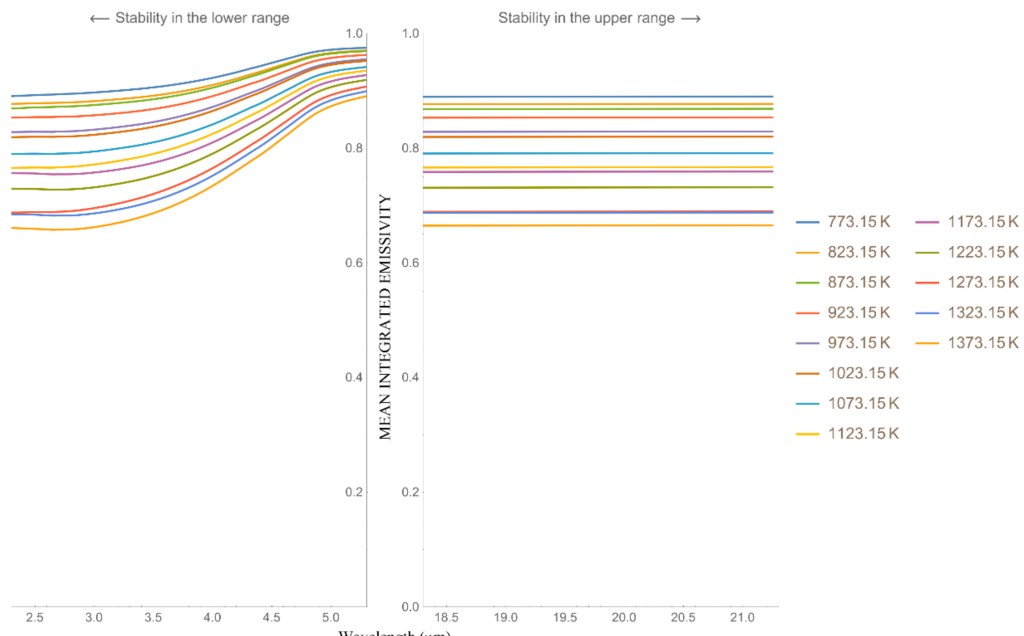

**Figure 6.** Behaviour of the upper and lower mean emissivity functions $\varepsilon_{T,-}(\lambda), \varepsilon_{T,+}(\lambda)$ as their range approaches the full available spectrum. The flattening of the curves indicates that the mean emissivity stabilises enough to allow us to approximate the full spectrum mean emissivity with the limiting value for each given temperature.

With this qualitative analysis supporting our assumptions, we approximate the full spectrum mean emissivity with it, and fit a quadratic polynomial for the available samples (Figure 7), giving us the mean emissivity of lava as a function of temperature as:

$$\varepsilon(T) \simeq \varepsilon_{[2.17,21]}(T) = 0.97672 + 0.00004\, T - 1.95062 \cdot 10^{-7}\, T^2 \tag{10}$$

Similar to the single channel emissivity functions, the effective trend is shown in Figure 7.

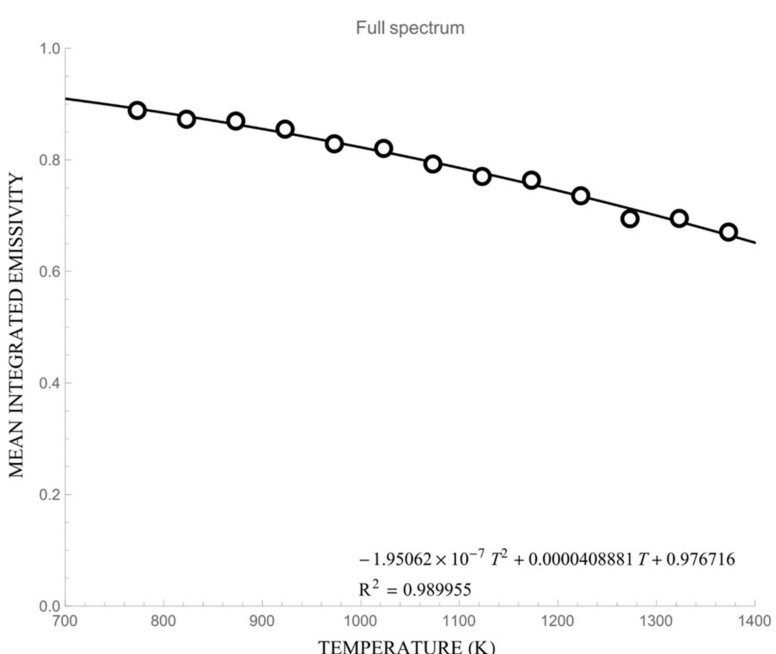

**Figure 7.** Emissivity–temperature trend at a range of temperatures (773–1373 K) for the full spectrum, with uncertainty <4%, as reported.

### 3.2. Spaceborne Data Results: Computation of Radiant Heat Flux

#### 3.2.1. Landsat 7-ETM+ Data

The spaceborne scene acquired during the 2001 Mt. Etna eruption (Figure 8) was used to isolate thermally anomalous pixels, corresponding to recorded radiances in the two ETM+ SWIR bands (Band 5 and 7), which were used to compute the radiant heat flux, following the method detailed in Section 2.3. Measured emissivity–temperature data in the SWIR (2.17–2.35 µm) region were used as input parameters (Table 2), which allocate specific emissivity in computation of radiant heat flux from ETM+ spaceborne data, using a 'thresholding' approach (Section 2.3). Considering that emissivity at SWIR 1 (1.55–1.75 µm) wavelengths was not measured due to the FTIR instrument limitations (expected to behave similarly to SWIR 2), assumed (extrapolated) maximum emissivity values ($\varepsilon_{max}$) for SWIR 1 were allocated for each temperature step (Table 2).

A total of 958 radiant pixels, extracted from the Landsat-7 scene are analysed here, using multicomponent emissivity-based allocation of specific emissivity values (Table 2) based on recorded spectral radiance (W m$^{-2}$sr$^{-1}$µm$^{-1}$) thresholds indicated in Figure 8 (bottom right). Here, emissivity of 0.84 was applied to 420 pixels, 0.83 to 370 pixels, and 0.82 to 168 pixels, whereas an emissivity of 0.81 and 0.80 was applied to none, as the highest recorded radiance in Band 7 for this scene was 16.5 W m$^{-2}$sr$^{-1}$µm$^{-1}$. Computation of radiant heat flux using the multicomponent emissivity approach for the scene on 5 August 2001 produced a value of 2.03 gigawatts (GW).

#### 3.2.2. MODIS Data

MODIS scenes acquired by Terra satellite over Mt Etna from 1 July to 30 August 2001 were processed to evaluate the contribution of the temperature-dependent emissivity to the estimation of the satellite-derived effusion rate. Thus, we solved the system of Equations (3)–(5) considering that the emissivity is a function of temperature and wavelength, according to Equations (7) and (8). In order to evaluate the per-pixel expected difference in the related radiant heat flux, i.e., the quantity we usually derive from satellite data to obtain an estimation of the effusion rate, we first solved the system by considering both a variable and a constant emissivity for a number of synthetic pixels. To build the synthetic pixels, we assumed for pc a range of variation from 0.0001 to 0.1, for pb between

0.8 and 0.99, and Tc between 400 and 800 K, fixed Tb equal to 300 K and Th to 1373 K, and computed the integrated radiance. We then selected about 4000 pixels giving admissible values (e.g., $ph = 1 - pb - pc > 0$), solved the system of equations (Equations (3)–(5)), and computed the difference between the radiant heat flux associated with the variable emissivity against that associated with a constant emissivity of 0.9. In computing the radiant heat flux, Equation (10) was applied to derive the emissivity.

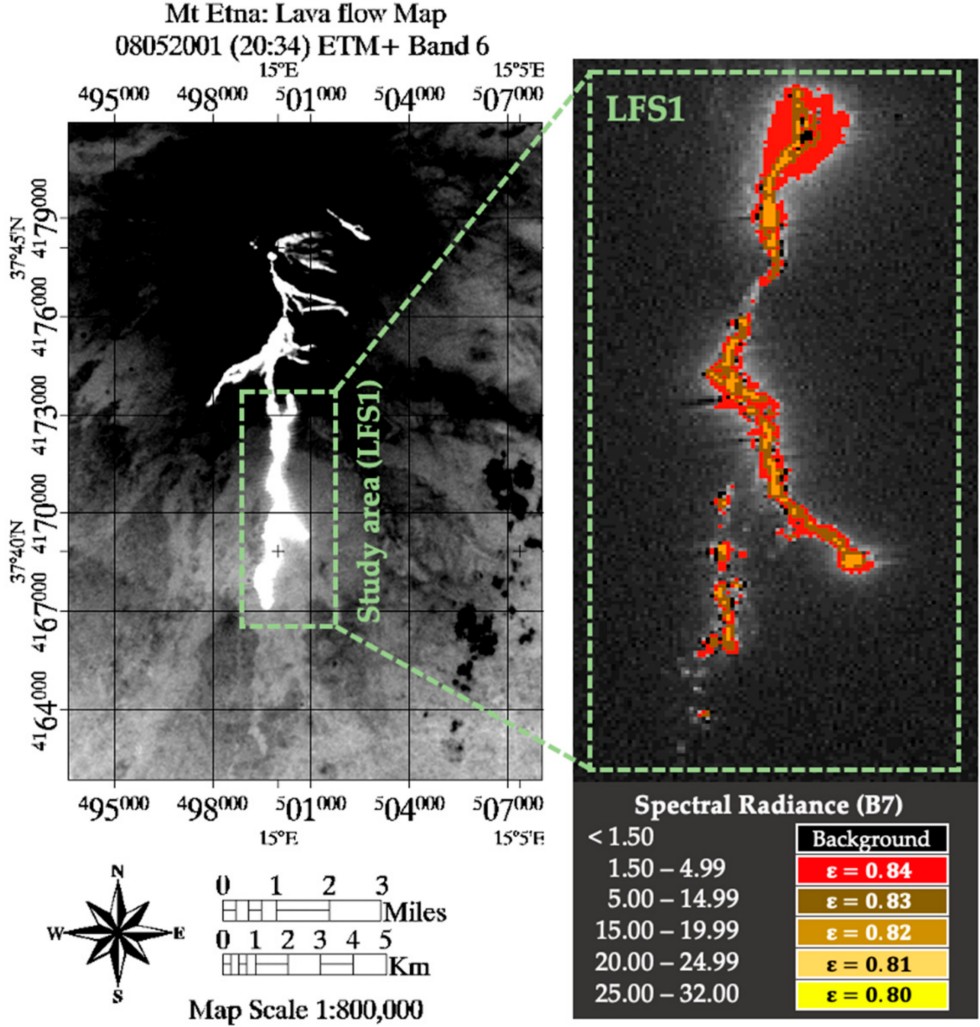

**Figure 8.** (**left**) Study area (LFS1) [7] indicated on ETM+ TIR (Band 6) image, used here for visual presentation purposes alone; (**right**) ETM+ scene, Band 7, acquired on 5 August 2001, showing all radiant pixels, within the high-temperature thermal anomaly, used for computation of radiant heat flux in this study (both SWIR Bands 5 and 7), using a multicomponent emissivity approach. The spectral radiance (W m$^{-2}$sr$^{-1}$μm$^{-1}$) 'thresholding' values used are shown in the table below the inset on the right.

**Table 2.** Emissivity in SWIR Bands at a range of temperatures (773–1373 K).

| Temperature (K) | 773 | 823 | 873 | 923 | 973 | 1023 | 1073 | 1123 | 1173 | 1223 | 1273 | 1323 | 1373 |
|---|---|---|---|---|---|---|---|---|---|---|---|---|---|
| $\varepsilon_{max}$ SWIR 1 | 0.805 | 0.804 | 0.837 | 0.867 | 0.857 | 0.846 | 0.825 | 0.805 | 0.825 | 0.791 | 0.736 | 0.755 | 0.753 |
| $\varepsilon_{measured}$ SWIR 2 | 0.805 | 0.804 | 0.837 | 0.867 | 0.857 | 0.846 | 0.825 | 0.805 | 0.825 | 0.791 | 0.736 | 0.755 | 0.753 |
| Error (Series) | 0.012 | 0.047 | 0.012 | 0.016 | 0.004 | 0.007 | 0.010 | 0.021 | 0.025 | 0.020 | 0.022 | 0.029 | 0.032 |
| Radiance SWIR1 | 2.14 | 4.50 | 8.10 | 14.3 | 23.0 | 34.8 | 50.96 | 70.71 | 101.4 | 131.1 | 161.8 | 214.6 | 267.3 |
| Radiance SWIR2 | 4.30 | 7.60 | 11.4 | 17.5 | 25.4 | 33.4 | 44.8 | 56.3 | 75.8 | 90.5 | 104.4 | 130.8 | 151.3 |

Figure 9 shows the histogram of the relative error between the variable emissivity and the constant pixels' radiant heat flux or radiative power (RP). The relative error is computed as:

$$\Delta \mathrm{RP} = 2\left(\frac{RP\ \varepsilon_{variable} - RP\ \varepsilon_{constant}}{RP\ \varepsilon_{variable} + RP\ \varepsilon_{constant}}\right) \tag{11}$$

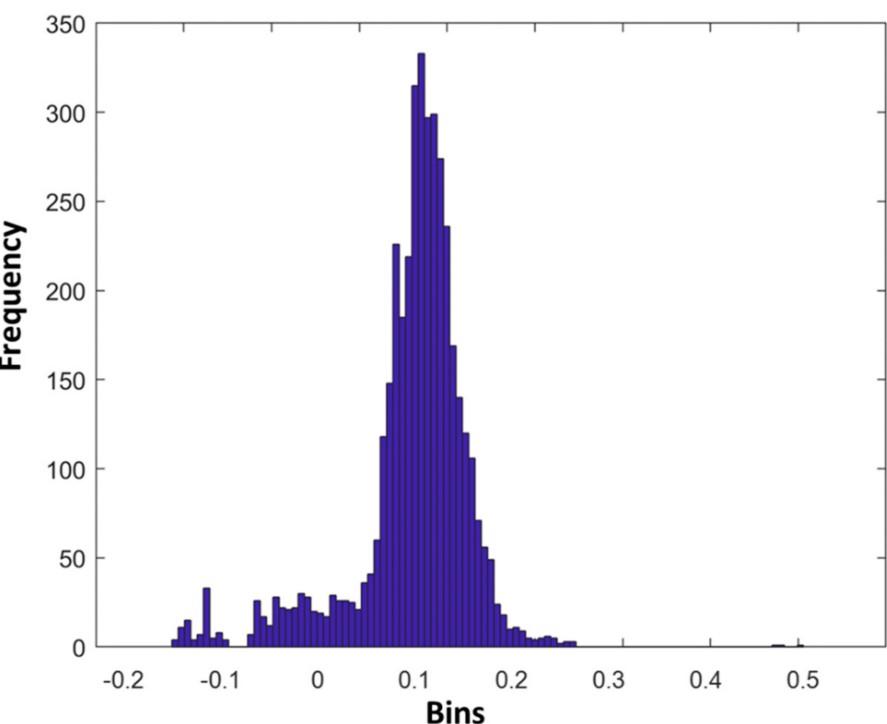

**Figure 9.** Histogram of the difference between the radiant heat flux for variable emissivity and constant emissivity computed for ~4000 theoretical pixels. The median value is 0.15 and the standard deviation is 0.13.

Generally, the variable emissivity produces greater radiant heat flux. The median value for the relative error is 0.15, whereas 0.08 is the 25th percentile and 0.23 the 75th percentile. Thus, for the range of values taken into account, we found the relative error is below 23% for 75% of the simulated pixels.

Regarding the real case study (i.e., the 2001 Mt. Etna eruption), we applied the HOT-SAT hotspot detection algorithm [40] and solved the system of equations (Equations (3)–(5)). Figure 10 shows the radiant heat flux values found for each thermally anomalous pixel, from 15 July to 30 August 2001. Results confirm the behaviour found for theoretical pixels, with the radiant heat flux derived using variable emissivity usually greater than the constant emissivity assumption. In this case, however, computing the same statistics on the difference between the radiant heat flux in case of variable emissivity and the case of constant emissivity, we found the median value is 0.06, whereas the 25th and the 75th percentiles are 0.014 and 0.161, respectively.

### 3.3. Lava Flow Modelling Results

Numerical modelling was performed using GPUFLOW [41,42], a physics-based model for the spatiotemporal evolution of lava flows, based on the Cellular Automaton paradigm, whose numerical stability, reliability, and accuracy has been assessed in previous sensitivity analyses [43,44]. The model has been successfully used to forecast different eruptive scenarios in various volcanic areas worldwide [5,45–48], and to assess lava flow hazards [48–50] and mitigate the associated risk [51–53].

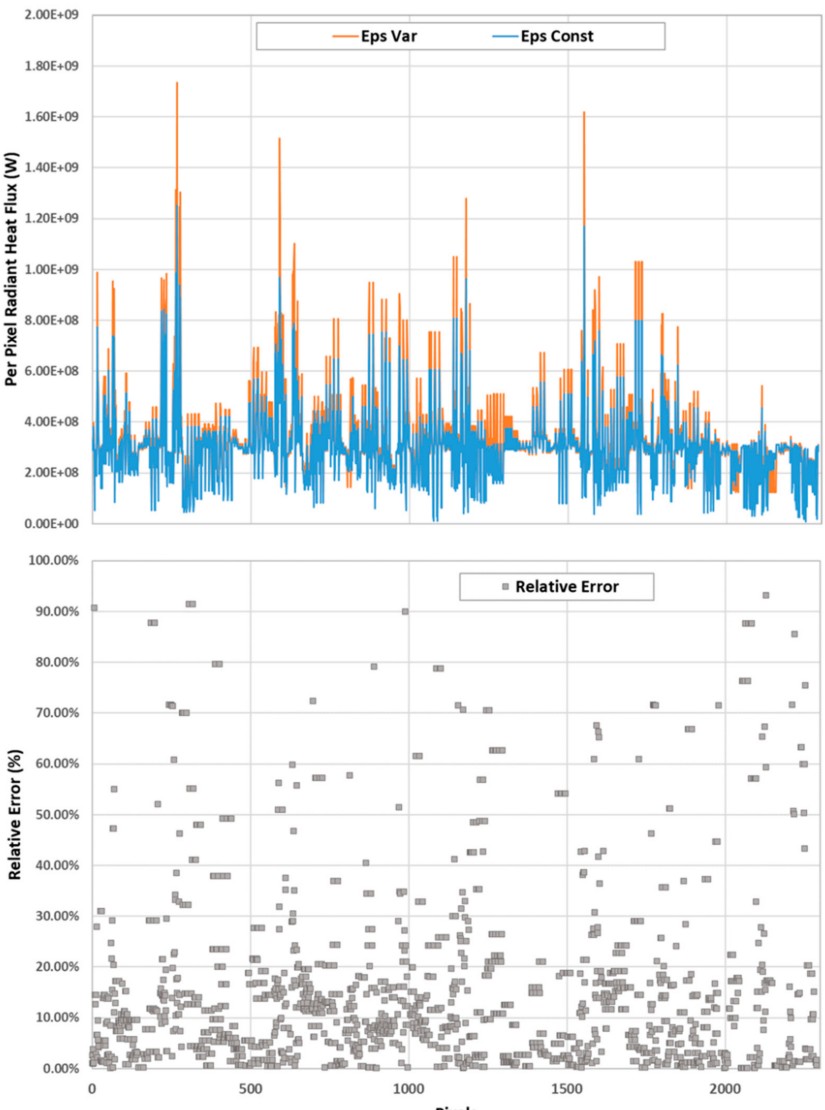

**Figure 10.** (**Top**)—Per pixel radiant heat flux for variable emissivity (orange) and constant emissivity (blue). (**Bottom**)—Relative error (grey squares) computed for MODIS images during 15 July–30 August 2001 at Mt. Etna.

To estimate the influence that the variation of emissivity has on the lava flow emplacement, we performed two synthetic lava flow simulations, both starting from a single vent located on a 20-degree inclined plane (i.e., the DEM, having a horizontal resolution of 5 m). For the rheological properties, we used the typical parameters of Etnean lava (i.e., 1360 and 1143 K as extrusion and solidification temperatures, respectively, 2600 kg m$^{-3}$ as density, and 0.02 wt% as water content).

The only input parameter that differs in the two synthetic simulations ('Simulation 1' and 'Simulation 2'), is the emissivity. In 'Simulation 1', GPUFLOW was executed using a constant emissivity of 0.90, which is a typical value for basaltic surfaces, such as polished and rough basalts at 0.90 and 0.95, respectively [1], whereas 'Simulation 2' uses the temperature–dependent emissivity equation (Equation (10)). The radiant heat flux values obtained for the 2001 Etna eruption using a constant and multicomponent emissivity were converted in the two TADRs reported in Figure 11. The TADR with constant emissivity, termed 'TADR 1' here, shows a peak on 21 July at 21:50 GMT of 34.0 m$^3$s$^{-1}$, whereas 'TADR 2' has a higher peak of 38.0 m$^3$s$^{-1}$. The cumulated volume calculated from 'TADR 2' is slightly higher than that of 'TADR 1' (26.4 and 23.4 million m$^3$ respectively).

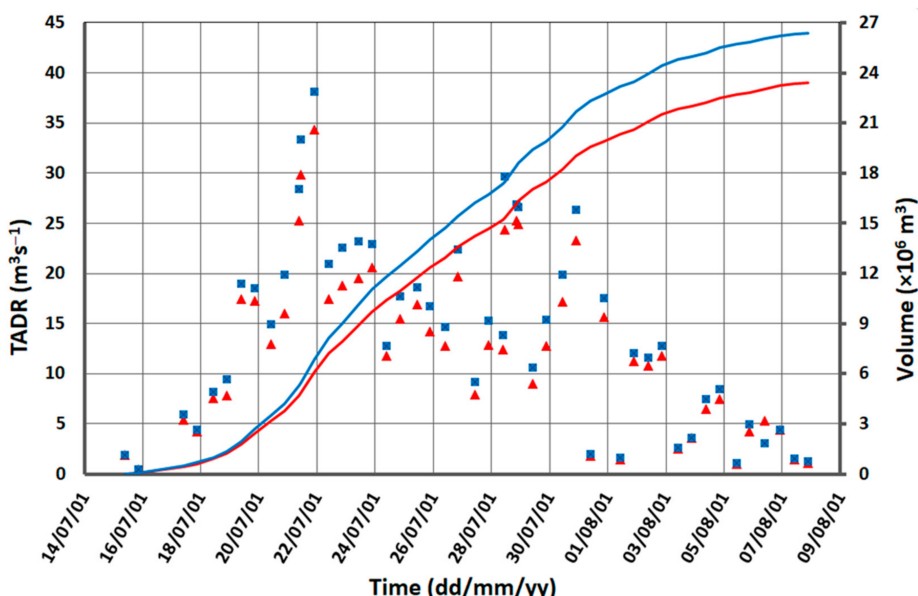

**Figure 11.** MODIS-derived TADR using a constant ('TADR 1', red triangles) and the multicomponent emissivity ('TADR 2', blue squares). The cumulative volumes are also reported (red curve for 'TADR 1', blue curve for 'TADR 2').

The final simulations are presented in Figure 12. The maximum length reached by the lava flow of 'Simulation 1' is 5.87 km, whereas for 'Simulation 2' it is 6.53 km. The differences are also evident in the maximum width (550 versus 570 m, respectively), in the area (2.09 versus 2.43 km$^2$, respectively), and in the maximum thickness (30 versus 28 m respectively).

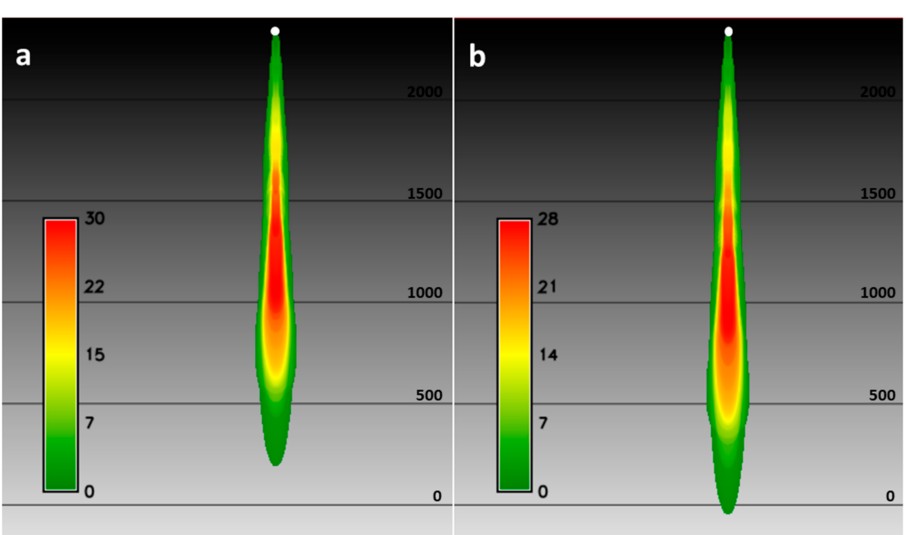

**Figure 12.** Final lava flow emplacements simulated by GPUFLOW on a 20-degree inclined DEM using (**a**) constant emissivity of 0.9 and (**b**) temperature-dependent emissivity. Colours represent the thickness of lava in meters, as reported in the legends. The white point indicates the flow start position, whereas the black lines represent the contours of altitude (500 m intervals, from 2000 m to 0 m).

## 4. Discussion

Spectral signatures (Figure 4) are consistent and relatively comparable with previous laboratory-based research of basaltic rocks in TIR (8.0–15.0 μm) at a low (~353 K) temperature [23]. However, our findings involve additional a high-to-very-high temperature range (773–1373 K) and shorter wavelengths (2.17–8.0 μm), which demonstrate and confirm that

emissivity of geological materials is not only dependent on the wavelength, but also the temperature [14,16,54].

Observed trends in the upper SWIR (2.17–2.35 μm) and MIR (3.0–5.50 μm) show a marked difference and more complex spectral shapes (i.e., lower emissivity and different shape signature), compared to the previous study of the same eruption (and same samples), in which reflectance data approach was employed [9].

Therefore, the FTIR results presented here (Figure 4) suggest that it is essential to assess the role and significance of emissivity, not only as a 'static' and uniform value relating to the solidified (cooled) surface, across all wavelengths and temperatures, but also taking account of its response to thermal gradient and the emissivity–temperature link, which were previously explored only at much lower temperatures and limited spectral resolution [9,13,16,23]. This approach allowed us to determine the emissivity variation with temperature change, in addition to the sensitivity of numerical lava flow modelling to the emissivity parameter. This also provoked further investigation into the role and impact of emissivity in lava flow dynamic modelling and hazard mitigation using spaceborne data, as demonstrated in this study.

In order to make emissivity a standard input parameter, and to develop a procedure for both spaceborne and modelling applications for Mt. Etna, the very-high temperature laboratory data we used, as presented here, appears to be most complete available. It covers the most appropriate temperature range for Mt. Etna ($\leq$1360 K) and nearly the full wavelength range used in remote sensing (i.e., SWIR, MIR, TIR).

The impact of the dependency of emissivity on temperature in applications is currently within the margin of errors introduced by the processing of the remote sensing data (e.g., the radiant heat flux curve usually has an uncertainty of $\pm$30%, [29]), and by the approximations in the modelling tools [43,44]. A more significant impact of the temperature-dependent emissivity can instead be expected in the development of more sophisticated and complete models for lava flow simulations, and in the development of more sophisticated remote sensing techniques.

In remote sensing, an area of interest that deserves a more in-depth study is the interaction between the emissivity (and its variability) and the TADR conversion constant. We can follow the argument that the conversion is affected by the viscosity of lava because lower viscosity leads to greater surface areas for the same mass flux rates [55]. On the other hand, the viscosity of lava has a strong dependency on temperature, with variations of two orders of magnitude between effusion and solidus temperature [31], where emissivity affects the surface heat loss, and lower emissivity leads to higher temperatures being maintained for a longer time, corresponding to lower viscosities. Therefore, with a 'dynamic experimentally deduced emissivity–temperature relation', it may be necessary to develop temperature-dependent TADR conversion functions that take into account how variable emissivity affects the change in viscosity during the flow.

To study the relationship between variable emissivity and the evolution of viscosity during the flow, the temperature-dependent emissivity relation (Equation (10)) may be integrated in more sophisticated physical–mathematical models for lava flow simulations, such as GPUSPH [56–58]. Running several simulations in GPUSPH, in controlled conditions, it will be possible to study both how the reduced heat loss affects the change in viscosity during the eruption, and the influence this has on the surface extension and ultimately the heat map, as it would be perceived by remote sensing instruments.

We found that the satellite-retrieved radiant heat flux, considering a temperature varying emissivity, is lower than what most studies used previously. In the case of the 2001 Mt. Etna eruption, most of the differences are below 16%, even if for some pixels this difference can be more than 90%. Results for theoretical pixels show a bigger discrepancy, with the relative error below 23% for the 75% of the simulated pixels.

The novel technique presented here, in which multicomponent emissivity is applied to SWIR ETM+ data, chosen for spaceborne analysis, provides a useful 'snap-shot', indicative of the current (instantaneous) state of activity, and cannot be used as a stand-alone approach.

Therefore, we use it here in tandem with MODIS data to complement the computed radiant heat flux range for the eruption and period analysed.

Furthermore, the computed radiant heat flux acquired on 5 August 2001, from two markedly different platforms, showed values of 2.03 GW for ETM+ (20:35 UTM) and 2.56 ± 1.09 GW MODIS (21:10 UTM) if we consider the hotspot MODIS overlapping the LSF1 flow (Figure 13). Despite spatial and temporal differences, the result produced by two different platforms agree and can be used to constrain the radiant heat flux range and uncertainty by combining available data and using the multicomponent emissivity approach. Moreover, the laboratory work reported here should be extended to include the lower SWIR wavelengths (e.g., 1.55–1.75 µm) for more accurate and complete radiant heat flux estimates. The effect of considering the variability in emissivity on Mt. Etna has a moderate but measurable impact on the forecasting of the emplacement (in the order of 10%, and up to 15% if considering different inclinations of the DEM [41,42]). This is partially due to the range of temperatures and viscosities of the lava on Mt. Etna. We expect the impact to be more significant in volcanoes where lava reaches higher temperatures and has lower viscosity (e.g., Kilauea, Hawaii; Fogo, Cape Verde; Fagradalsfjall, Iceland; Nyiragongo, DR Congo), due to the combined effect of higher emissivity variation and the higher distances covered with the resulting reduced cooling rate [16].

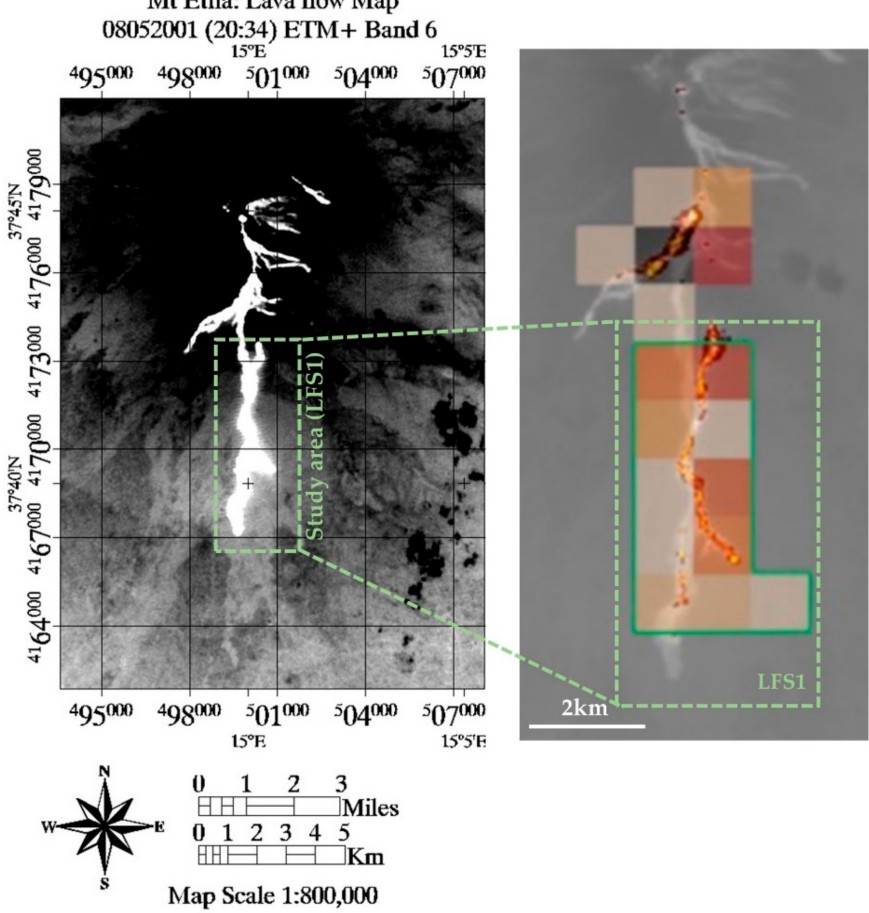

**Figure 13.** (**left**) Study area (LFS1) [7] indicated on a ETM+ TIR (Band 6) image, used here for visual presentation purposes alone; (**right**) hotspot pixels derived from MODIS data, acquired on 5 August at 21:10 UTC, superimposed on the Landsat 7, Band 7 data acquired on 5 August at 20:34 UTC, as SWIR Bands (5 and 7) were used for ETM+ computation of radiant heat flux. MODIS pixel colours are related to the value of radiant heat flux (e.g., bright red—high values, pale red—low values). The dashed green line indicates the extent of LFS1 study area, whereas the solid green border line marks MODIS pixels relating to ETM+ radiant pixel area in Band 7.

The conversion of the two radiant heat flux results (using a constant and multicomponent emissivity) produced two TADR curves for the lava flow simulations, with peaks that differ by 4.0 $m^3 s^{-1}$, determining a deficit of ~14% in the total DRE volume obtained using a constant emissivity (Figure 11). This has a direct consequence in the lava flow modelling, as demonstrated by the two synthetic GPUFLOW simulations (Figure 12). Indeed, the distribution of 'Simulation 1' is more 'compact' than that of 'Simulation 2', having a narrower width (~4% less) and a shorter length (~11% less). This is mainly due to the difference in the TADR peaks. The length change (~11%) is similar to that of a recent study that investigated the impact of multicomponent emissivity on lava flow modelling (~7%) [16]. Conversely, the difference in volume has more impact in the area and thickness of the two simulated lava flows, causing a dissimilarity of ~16% in the final area and of ~7% in the maximum thickness.

## 5. Conclusions

Volcanologists and modellers have for many years relied on coarse spatial resolution (≥1 km) spaceborne data in MIR and TIR, overlooking the impact that input parameters, such as emissivity, can have on monitoring active volcanoes. This may be driven both by a lack of reliable information on emissivity's behaviour with temperature and by the dynamic nature of volcanic hazards, favouring a higher repeat interval (temporal resolution) over the greater detail (spatial resolution).

Current operational satellite-based volcano monitoring approaches, using moderate-to-high temporal resolution data alone, would benefit from combining high-spatial resolution data (e.g., Landsat series and/or MSI for Sentinel 2), where available. This would improve the accuracy of operational and tactical volcanic crises management. In addition to this, using appropriate input parameters, such as emissivity values that reflect variation with temperature (multicomponent emissivity), would provide improved information to constrain thermal phenomena, such as lava flow lengths, and estimates related to volcanic radiant heat flux, in addition to modelling applications [12].

Solitary, high-spatial resolution image data, with FTIR emissivity data in the upper SWIR region, were used in this study to assess potential uncertainty in the radiant heat flux calculated from moderate-to-high temporal resolution image data (MODIS). It is, however, beneficial to exploit the enormous amount of currently available high-temporal resolution baseline data to quantify the natural variability of the volcanic systems under investigation, as solitary 'snap-shot' data alone cannot produce the temporal detail needed to track hourly changes in activity at individual ongoing eruptions.

The impact that temperature-dependent emissivity has in the modelling and simulation of lava flow dynamics in a digital 3D environment is twofold. First, for a given mass flux rate, the lower emissivity at higher temperature can lead to emplacement differences with respect to simulations with a constant emissivity, due to the temperature-dependent viscosity of lava. Second, particularly in nowcasting applications, variable emissivity affects the mass flux rate estimation itself. The combined effect of these aspects can lead to differences in space of between 10% and 20% in the emplacement of the flow. A more detailed analysis of the direct and indirect impact of the influence of the variable emissivity, with more sophisticated physical models (e.g., GPUSPH), may shed further insights on the relative importance of the relation between emissivity and temperature, both for remote sensing and modelling applications.

The results in this study show that emissivity is both temperature and wavelength dependent. Measured emissivity increases non-linearly with temperature decrease (cooling), exhibiting significant variations above 900 K with values considerably lower than the typically assumed 0.95 (especially in MIR). This new evidence has a measurable impact on the computation of radiant heat flux from spaceborne data, and on modelling of lava flow 'distance-to-run' simulations. Furnished with improved input parameters (multicomponent emissivity), the novel approach developed here can improve the reliability of results. This

will lead to more complex and realistic spaceborne multi-platform, multi-payload volcano monitoring systems for hazard assessment.

**Author Contributions:** F.F. and N.R. led the conception and design of the work. N.R., J.O.T., G.B., G.G. and A.C. acquired the data. N.R., J.O.T., G.B., G.G., A.C., H.R., M.S.R. and F.F. were responsible for analysis, modelling, and interpretation of data. All authors have read and agreed to the published version of the manuscript.

**Funding:** The Open University, Milton Keynes, UK provided studentship funding for N.R.

**Acknowledgments:** Thanks are due to National Aeronautics and Space Administration (NASA) for MODIS data (modis.gsfc.nasa.gov, accessed on 20 February 2022) data. Special thanks to S. Eriksen for support and guidance relating to data processing.

**Conflicts of Interest:** The authors declare no conflict of interest. The funders had no role in the design of the study; in the collection, analyses, or interpretation of data; in the writing of the manuscript, or in the decision to publish the results.

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
