# Peer review of "The Impact of Dynamic Emissivity–Temperature Trends on Spaceborne Data: Applications to the 2001 Mount Etna Eruption"

_remotesensing, doi:10.3390/rs14071641_

Round 1
Reviewer 1 Report
Dear Authors,
Please find below a set of comments and remarks I would kindly ask you to revisit in your manuscript as I believe
that they would improve the overall merit of your work. The paper is well written and the majority of comments are minor, however a couple of them are rather important and should be taken care of before the manuscript be considered for publication. If all is taken care of I would be happy to recommend your work for publication
General comments:
The data presented are from 2001. There is no strong motivation in the manuscript highlighting the importance of this eruption. And why did it take 20 years for you to perform this analysis? Some general information on the eruption would help along these lines. Please consider strengthening this part early in the manuscript.
Also, the discussion should put more emphasis on the cross combination of the remote sensing and lab-based technique.
Despite time delaying, your approach provides the means for a "best practice" scenario when accurate data required. Please consider adding some text along these lines.
In addition:
l.141
is the blackbody source standardized? If yes, please mention its standard data, protocol or similar info.
l.191 and elsewhere
It would be better to avoid the term "law" (with our withour quotes) for the experimentally deduced emissivity-temperature relation. In the context of the paper and to the extent this correlation is established, there is no basis for a "law".
l.193 and for the full paragraph
please fix tenses, as almost all sentences use the present tense. The paragraph sounds like a recipe, not a scientific methodology
l.217
MATLAB is licensed software. Please acknowledge the license (private, corporate or institutional).
l.229 and elsewhere
please put URLs listed in the Reference list
l.271, 274, 277 and 279
what defines the number of decimal digits in the produced coefficients? If we take into account the quoted relative uncertainties (e.g. 0.07%) the use of 6 or 7 digitis is completely meaningless. Please take care of significant digits in these equations. what are the uncertainties of the coefficients as they are deduced from the fit?
l.282 (fig. 5)
the text is hard to read. Please consider splitting the image in a 2x2 arrangement of subfigures
l.286 and the whole paragraph
I find the discussion about 'stable' behavior of the function and the whole treatment a little vague. How much is 'stable'? how much is 'small' for the change? The quantification in terms of the discrete derivable is not given, with the exception of a maximum value of 5% for one value of the wavelength. The lower range (fig. 6 - left) has a change of ~15-20%. Please add some comments to clarify the situation.
l.315
see comment about significant digits earlier. The uncertainty is traceable to eq. 10 (or the opposite)
l.316
The trend is CLEARLY not linear, either in visual or in arithmetical grounds. Please remove the statements
l.317 (fig. 7)
what is the uncertainty in the data points? Please add a comment in the caption
l.328
what is the 'recipe' to assume the values of emissivity? extrapolation? please mention that explicitly in the text
l.370
please provide a reference for the algorithm
l.392-393
there is a spurious blank line between bullets ii and iii
l.474
"below 16%"
l. 482 fig. 13
please magnify the figure to assist readability
l.504 and the whole paragraph.
How strong is the dependence of the simulated results on the initial inclination parameter or other geomorphological parameters used? A comments is necessary. e.g. if you change the 20 deg what and how much does it change? Have you run such alternative scenarios? And do the initial parameters reflect the realistic features of Mt. Etna? A comment would ensure the reader on the merit of the method.
Author Response
Please see our correction notes attached
Manuscript ID: remotesensing-1624693
Title: The impact of dynamic emissivity-temperature trends on spaceborne data
of the 2001 Mount Etna eruption
We would like to take this opportunity to thank you for your feedback and guidance improving our manuscript.
Best wishes

Reviewer 2 Report
Dear Authors,
I am very appreciated to read your interesting manuscript presenting the dynamic emissivity values for the 2001 eruption of Mt. Etna. I have included my comments to the annotated pdf where you can find some suggestions mostly regarding the clarification of your sentences. It would be really great if you could rephrase the sentences and go through the plain language. Except for that, I think your manuscript seems promising to be published after a minor revision.
Best regards

Author Response

(The authors gave the same response as above.)

Reviewer 3 Report
This paper explores how emissivity changes with temperature and its influence on radiant heat flux estimations and lava flow simulations. This manuscript is interesting, well suited for this journal, and well written. However, there are several minor but important issues that should be addressed before being accepted for publication:
- The title is not accurate. The dynamic emissivity-temperature trend does not impact spaceborne data, but the interpretation or processing of those data. I suggest re-writing the title taking this into account.
- Lines 29 and 30. Those differences in radiant heat flux (median and percentiles) must have units. What are the units?
- Lines 46-48. This sentence is long and not easy to understand. Please simplify this sentence.
- Line 53. “the” missing before “last two decades”.
- Lines 142-143. Why were blackbody spectra acquired at 50K higher and lower than the expected sample temperature, and not at the sample temperature? Please, add a sentence to justify this.
- Line 181-182 – and Section 3.1.2. I don’t see why a “stable” mean emissivity within the range analyzed justifies or not approximating the mean emissivity to the mean emissivity over the available range. Further justification is needed to use that approximation. Or if there is no better justification than the “stability” analysis, then I suggest removing that analysis from the paper and specifying that this is a necessary assumption because of the limited range of the instruments.
- Line 218. What is the influence of these assumptions in the final results? These need to be assessed since that may be a source of uncertainty.
- Line 252 (Fig. 4). To foster this message, I encourage adding a third panel to show emissivity vs temperature for a few different wavelengths.
- Lines 254-256. Where are those features? Are they shown in Fig. 4? Please, clarify.
- Equations (6)-(10). More robust analysis of uncertainties should be done. I don’t find providing maximum relative error, median, and standard deviation particularly useful in this context. Instead, please provide the uncertainty of the fitting parameters, so future users of these equations can estimate the uncertainty of the emissivity for their particular scenario.
- Line 302 and 303. The units of those numbers are missing.
so I recommend publication after addressing several minor but important comments that I detail below:
- Page 4, lines 36-37. I don’t see the increase of CL with frequency in Fig. 4a.
- This manuscript is plenty of abbreviations and definitions. Please, add a summary table with the meaning of the abbreviations and all the definitions needed.
- Page 5, lines 6-8. I miss a figure with temperature, wind speed and direction, snowfall, and rain that shows the lack of correlation with BB.
- Page 2, line 6. “… periodic pressure oscillations [23]”. This is unclear because all the mechanisms cited reflect periodic pressure oscillations somehow. For example, replace “periodic pressure oscillations [23]” with “periodic pressure perturbations triggered by permeable gas flow [23]”.
- Recent papers on other data-reduction approaches that I suggest to cite:
https://www.mdpi.com/2076-3263/10/4/142
https://earth-planets-space.springeropen.com/articles/10.1186/s40623-021-01506-0
https://agupubs.onlinelibrary.wiley.com/doi/full/10.1029/2019JB018980
Author Response

(The authors gave the same response as above.)
